# A Semi-Empirical Damage Model of Helankou Rocks Based on Acoustic Emission

**DOI:** 10.3390/ma16114001

**Published:** 2023-05-26

**Authors:** Youzhen Yang, Qingqing Lin, Hailong Ma, Jahanzaib Israr, Wei Liu, Yishen Zhao, Wenguo Ma, Gang Zhang, Hongbo Li

**Affiliations:** 1School of Physics and Electronic-Electrical Engineering, Ningxia University, Yinchuan 750021, China; lingqingqing13@163.com (Q.L.); jisrar@uet.edu.pk (J.I.); liuwei_990315@163.com (W.L.); zhao18749@163.com (Y.Z.); mwg@nxu.edu.cn (W.M.); zhanggang@nxu.edu.cn (G.Z.); 2Institute of Solid Mechanics, Ningxia University, Yinchuan 750021, China; 3Institute of Ethnic Preparatory Education, Ningxia University, Yinchuan 750021, China; mhl_323@163.com; 4Department of Civil Engineering, University of Engineering and Technology, Lahore 54890, Pakistan; 5College of Civil and Hydraulic Engineering, Ningxia University, Yinchuan 750021, China; 13469578416@nxu.edu.cn

**Keywords:** weathering, dry-wet, freeze-thaw, acoustic emission, damage model, Helankou rock, semi-empirical

## Abstract

The Helankou rock as the relics carrier in Ningxia, China, have been suffering from serious weathering caused by variable environmental conditions. To study the freeze-thaw damage characteristics of Helankou relics carrier rocks, three dry-wet conditions (i.e., drying, pH = 2 and pH = 7) together with freeze-thaw experiments have been carried out at 0, 10, 20, 30, and 40 cycles. Additionally, a series of triaxial compression tests have been carried out at four different cell pressures of 4 MPa, 8 MPa, 16 MPa, and 32 MPa in tandem with a non-destructive acoustic emission technique. Subsequently, the rock damage variables were identified based on elastic modulus and acoustic emission ringing counts. It has been revealed that the acoustic emission positioning points reflected that the cracks would be concentrated near the surface of main fracture at higher cell pressures. Notably, the rock samples at 0 freeze-thaw cycles failed in pure shear. However, both shear slip and extension along the tensile cracks were observed at 20 freeze-thaw cycles, while tensile-oblique shear failure occurred at 40 freeze-thaw cycles. Not surprisingly, the decreasing order of deterioration inside the rock was observed to be (drying group) > (pH = 7 group) > (pH = 2 group). The peak values of damage variables in these three groups were also found to be consistent with the deterioration trend observed under freeze-thaw cycles. Finally, the semi-empirical damage model could rigorously ascertain stress and deformation behavior of rock samples, thus providing theoretical basis to establish a protection framework for Helankou relics.

## 1. Introduction

Helankou is in the middle of Helan Mountains in Ningxia, as shown in Figure 1a, where the traces of human life from 3000 to 10,000 years ago have been preserved. Notably, these have been inscribed on the United Nation’s unofficial list of World Heritage Sites since 1997 and on China’s National Dual Natural and Cultural Heritage Tentative List since 2006 [1]. However, the relics have been suffering from serious weathering damages lately, including powder peeling, crack development, and local collapsing (Figure 1), and are now endangered to disappear. The progression of the damage is strongly related to the local climate changes, such as precipitation and temperature. For instance, the annual precipitation nears 400 mm in most areas of Ningxia, while the snow covering the top of mountains would melt to form a perennial stream, thereby creating a local microclimate. In recent years, the average monthly maximum and minimum temperatures have been 29.9 °C in July and −12.8 °C in January, respectively. In addition, according to the relevant records, the historical highest temperature reached 39 °C and lowest −31 °C. The monthly average relative humidity is 40~66% [2]. This long-term temperature and humidity cycling would cause severe deterioration that would aggravate further with acidic pollution such as smog, i.e., a complex mix of industrial smoke and natural fog [3]. Thus, in order to protect the Helankou relics, it would be crucial to evaluate the damage characteristics of carrier rock materials subjected to both dry-wet (D-W) and freeze-thaw (F-T) cycles through a robust laboratory program.

Thus far, in order to study the problem of rock weathering caused by environment [5], the rock damage characteristics under D-W and F-T conditions applying compression tests have been investigated by several researchers. For example, Khanlari et al. [6] reported variations of uniaxial compressive strength, wave velocity, and elastic modulus of rock due to F-T cycling. Similarly, Jia et al. [7] analyzed the destruction process and evolution law of sandstone pore structure with F-T cycling. Gratchev et al. [8] studied the effect of D-W cycle on the strength and fracture development of two kinds of hard rock. Furthermore, Zhang et al. [9] analyzed the variation of peak stress, residual strength, and elastic modulus of red sandstone under F-T and triaxial compression conditions. Mousavi et al. [10] reported the effects of F-T on rock damage under triaxial compression tests. Liao et al. [11] analyzed the variation of elastic modulus and residual strength with F-T and D-W cycles under triaxial compression. Lately, Zhou et al. [12] examined the effects of temperature changes on mechanical property of a rock under triaxial compression loading. In essence, the damage mechanism of rocks has been extensively examined at macrolevel, while little has been investigated on both crack localization and propagation processes.

Of late, many new technologies are applied in studying the rock damage characteristic, such as computed tomography (CT) [13], nuclear magnetic resonance [14], etc. Among them, acoustic emission (AE) is a novel technique that could effectively monitor both initiation and propagation of microcracks inside the rock material [15,16]. Thus far, it has been widely applied in medical treatment [17,18], alloys and metals [19,20], geotechnical and geological engineering processes [21,22], and gradually adopted in non-destructive testing of engineering structures [23,24]. For instance, Liu et al. [25] used AE techniques to analyze the microscopic damage evolution of loaded sandstone under the F-T cycle. Tang et al. [26] found that the ringing counts increased gradually in the loading stage, while the same increased manifold during the unloading stages. Several researchers have carried out studies on F-T damage characteristics of rocks and changes of AE parameters. For instance, Yang et al. [27] reported that uniaxial AE signal of limestone remained active until the characteristic of local high density released as the F-T cycles increased. 

For exploring the damage evolution further, various damage models have been established so far. For example, Fang et al. [28] proposed a universal damage model based on elastic modulus and peak stress under loading and F-T conditions. Later, Zhang et al. [29] defined damage variable by elastic modulus considering the heterogeneity of rock at mesoscopic level. Additionally, Liu et al. [30] established the damage model of coal-rock under uniaxial compression based on AE ringing counts and proposed a damage evolution curve. Liu et al. [31] presented the damage variable, which could rigorously reflect the evolution process involving the tensile damage of sandstone based on F-T cycles and cumulative ringing counts. In a previous study by these authors [32], damage variable had been defined across scales under F-T condition but without D-W environment, which has an impact on deterioration of rock materials. Jiang et al. [33] proposed a damage constitutive model simulating the entire deformation process of rock subjected to F-T cycles. Zhao et al. [34] proposed an elastoplastic damage model based on the deformation features of the frozen sandstone. Notably, the damage variables were established based on either a macroscopic quantity, such as elastic modulus, or a microscopic quantity, such as AE ringing count. However, damage has been rarely characterized by a combination of two or more characteristics leading to the failure based on cross-scale definition of the damage variable. In this study, the authors adopted a semi-empirical observational approach, whereby a new damage variable is defined based on elastic modulus and AE ringing counts to capture F-T cycle-induced damage mechanism of Helan Mountain relic carrier rocks.

Thus, the current study envisioned to characterize the damage based on multiple controlling factors for the sake of enhanced rigor and accuracy. Accordingly, a comprehensive laboratory program involving both geotechnical and geo-environmental tests has been designed, whereby F-T tests under three environmental conditions and triaxial compression tests under four cell pressures have been carried out with non-destructive AE testing. The variations of elastic modulus, peak stress, and residual stress under different environmental and loading conditions have been examined. Based on AE ringing counts, cumulative energy counts, and AE positioning points, initiation and propagation of cracks could be analyzed. Using both macro- and microcharacteristics of rock deformation, a new damage model based on elastic modulus and AE ringing counts has been presented. Nevertheless, the current results could significantly enhance our understanding of the damage characteristics of Helankou rocks that would form basis for establishing a robust protection framework for Helankou relics.

## 2. Laboratory Program

### 2.1. Specimens Preparation

In this study, fresh and intact rock blocks were chosen as the research objects from the Helankou rock relics. In essence, the mineral composition of the rock samples mainly consists of quartz (40%), feldspar (22%), calcite (18%), and clay minerals (20%). The angular-subangular quartz with a particle size of 0.15–0.9 mm, subangular feldspar with a particle size of 0.18–0.65 mm, and evenly distributed rock debris of 0.2–0.55 mm in the rock specimens. It is noteworthy that 100 mm cylindrical samples with diameter of 50 mm were extracted according to ISRM [35]. To ensure uniformity, the rock samples exhibiting similar ultrasonic wave velocity and bearing no obvious surface defects were selected for further testing. Subsequently, a total of 60 samples were finalized and further subdivided into three test groups; namely, (1) drying group, (2) pH = 2 group, and (3) pH = 7 group, labeled as G, 2, and 7, respectively. As Figure 2 shows, the freeze-thaw (F-T) cycles were set to 0, 10, 20, 30, and 40, labeled as A, B, C, D, and E, respectively. After F-T cycling, different magnitudes of the cell pressures were applied i.e., 4 MPa, 8 MPa, 16 MPa, and 32 MPa, labeled as 1, 2, 3, and 4, respectively.

### 2.2. Test Equipment and Scheme

It is noteworthy that the experimental parameters examined in this study have been selected based on the observations reported by the recent studies [8,36]. For brevity, drying groups were set up to study the effects of temperature changes (i.e., maximum temperature difference) on rocks, pH = 7 groups were examined for quantifying water and temperature changes (e.g., rocks immersed in streams for long periods of time are affected by water and temperature changes), and pH = 2 groups were considered for the effects of acid rain on rock materials [37].

In this study, the programmable high-low temperature device TMS9012-80 manufactured by Zhejiang Tomos Technology Company Limited in Hangzhou, China, shown in the lower left inset of Figure 3, was adopted to perform the F-T cycling tests. For completeness, the temperature range was −31 °C to 40 °C in this work, with an accuracy of ±0.5 °C. Similarly, TFD-1000G rock mechanics, manufactured by Changchun Keyi Test Instrument Company Limited in Changchun, China, was used to conduct triaxial compression (see lower right inset, Figure 3). The triaxial compression machine could apply a maximum axial load of 1000 kN, with an accuracy of ±1%, wherein the oil pressure technique was used to apply target cell pressures, and the specimens were axially loaded until failure. The location arrangement of AE sensors is shown in the lower right inset of Figure 3, whereby six sensors were arranged in two parallel rows of three sensors each. The upper three sensors were numbered from No. 1 to 3, while those below were numbered from 4 to 6. Notably, both sets of sensors were arranged counterclockwise and 20 mm away from the top of the sample at an angle of 120° from each other.

As Figure 3 shows, to replicate field environmental conditions of Helankou relics to accurately capture the rock damage in the laboratory, the following steps were taken:Each rock specimen was saturated by vacuum evacuation for at least 12 h. Subsequently, the saturated specimens were weighed and placed in an oven at 105 °C for a minimum of 12 h, and then cooled down to the room temperature.Rock samples group with pH = 2 and pH = 7 was placed in concentrated sulfuric acid at pH = 2 and distilled water for at least 12 h, respectively.All specimens were subjected to cycles of high-low temperatures, which varied between −31 °C for 2 to 4 h, and then it increased to 40 °C for 2 to 4 h, as shown in the upper right inset of Figure 3. Notably, 0, 10, 20, 30, and 40 freeze-thaw (F-T) cycles were applied in this study.After F-T cycling, the rock samples of pH = 2 and pH = 7 groups were dried for 12 h and cooled down. The two groups of rock samples were soaked and dried before and after each cycle.Each specimen was subjected to triaxial compression testing under 4 MPa, 8 MPa, 16 MPa, and 32 MPa cell pressures, and the acoustic emission (AE) non-destructive testing.

## 3. Results and Discussion

### 3.1. Analysis of Elastic Modulus Variation

The elastic moduli of rock specimens could be obtained through a series of triaxial compression tests. In this study, the $E_{N_{t}}$ values have been deduced from the stress–strain curves obtained from triaxial compression tests results of samples subjected to $N_{t}$ number of F-T cycles under different cell pressures ($\sigma_{c}$) from 4 MPa to 32 MPa. The resulting values of $E_{N_{t}}$ could be correlated with both $\sigma_{c}$ and F-T cycles ($N_{t}$) through a multivariate linear function, as follows:(1)$$E_{N_{t}}=a_{1}+b_{1}N_{t}+c_{1}\sigma_{c}$$
where $E_{N_{t}}$ is the elastic modulus after $N_{t}$ freeze-thaw (F-T) cycles; $a_{1}$, $b_{1}$, and $c_{1}$ are model parameters for elastic moduli, summarized in Table 1.

Figure 4 shows experimentally observed values of $E_{N_{t}}$ plotted against the F-T cycles for all three tested groups (i.e., drying, pH = 2, and pH = 7). Not surprisingly, the $E_{N_{t}}$ are fully consistent with the proposed fitting curves obtained from Equation (1). Furthermore, regardless of the tested group (i.e., drying, pH = 2, and pH = 7), the elastic modulus decreases with the increasing number of F-T cycles, while the same would increase with the increasing $\sigma_{c}$ values. Notably, at a given F-T cycle and cell pressure, the magnitude of $E_{N_{t}}$ follows the order (from small to large): drying group < pH = 7 group < pH = 2 group. For instance, at $N_{t}$ = 40 and $\sigma_{c}$ = 32 MPa, the reductions in the magnitude of elastic modulus of the drying group, pH = 2 group, and pH = 7 group are recorded as 6.05 GPa, 18.73 GPa, and 13.49 GPa, respectively. In addition, the mass loss of rock samples in the pH = 2 group was significantly higher than that in the other two groups. After 40 F-T cycles, the mass loss rates of the dry group, the pH = 2 group, and the pH = 7 group are 0.0564%, 1.2800%, and 0.0767%, respectively.

### 3.2. Analysis of Residual Stress and Peak Stress Variations

#### 3.2.1. Effects of D-W and F-T Conditions

Figure 5 presents the experimental results of both residual stress and peak stress magnitudes for all three groups tested in this study. The residual stress in the pH = 2 group after 0, 10, 20, 30, and 40 F-T cycles is 155 MPa, 74 MPa, 50 MPa, 42 MPa, and 9 MPa, respectively, and the peak stress is 473 MPa, 364 MPa, 332 MPa, 261 MPa, and 150 MPa, respectively. As can be seen that both residual stress and peak stress are gradually decreasing with the increasing F-T cycles. This shows that during the F-T process, the cementation continues to diminish, while the number of pores and fissures continues to evolve. As a result, the pore spaces between the constituent particles of a rock continue to expand, thus causing marked reduction in the overall rock strength. Nevertheless, the residual stress and peak stress could be fitted by a non-linear function:(2)$$\sigma_{r}{=a}_{2}+b_{2}N_{t}{}^{c_{2}}$$
(3)$$\sigma_{b}{=a}_{3}+b_{3}N_{t}{}^{c_{3}}$$
where $N_{t}$ is the number of F-T cycles, $\sigma_{r}$ is the residual stress and $\sigma_{b}$ is the peak stress, $a_{2}$, $b_{2}$, $c_{2}$, $a_{3}$, $b_{3}$, and $c_{3}$ are the model parameters given in Table 1, wherein the fitting coefficients of all three groups are close to 1.

The model predictions have also been plotted against the experimental values in Figure 5, where experimental values plot closer to the model predictions, thus showing good agreement. Both peak and residual stresses show good non-linear correlation with F-T cycles.

#### 3.2.2. Effect of Cell Pressure

The residual stress in the drying group under cell pressure of 4 MPa, 8 MPa, 16 MPa, and 32 MPa have been observed as 19 MPa, 37 MPa, 86 MPa, and 132 MPa, respectively, and the peak stress magnitudes as 242 MPa, 249 MPa, 336 MPa, and 411 MPa, respectively. As can be distinguished, the residual stress and peak stress gradually increase as the cell pressure increases. Notably, both peak and residual stresses under a given F-T cycle show non-linear correlation with the varying cell pressures, while the values of the fitting coefficient of each group are closer to 1. In essence, the anticipated correlations between residual stress, peak stress, and cell pressure read:(4)$$\sigma_{r}{=a}_{4}{+b}_{4}\sigma_{c}{}^{c_{4}}$$
(5)$$\sigma_{b}{=a}_{5}{+b}_{5}\sigma_{c}{}^{c_{5}}$$
where $\sigma_{c}$ is the cell pressure, and $a_{4}$, $b_{4}$, $c_{4}$, $a_{5}$, $b_{5}$, and $c_{5}$ are the model parameters (see Table 1). The fitting curves and experimental values under various cell pressures are shown in Figure 5, wherein both values predicted by the fitting curves and those determined experimentally are fully consistent.

### 3.3. Chemically Induced Variations of AE Parameters

The non-destructive testing approach involving the use of acoustic emission (AE) ringing counts and cumulative energy (CE) counts have also been adopted to monitor the rock damage in this study. For cell pressures ranging between 4 MPa and 32 MPa, time histories of AE, CE, and Axial (deviator) stress for 0 to 40 F-T cycles have been reported in Figure 6, Figure 7 and Figure 8 for the drying group, pH = 2 group, and pH = 7 group, respectively. During triaxial testing, the fracture of rock sample occurred instantly (i.e., brittle). At this moment, both stress and strain values changed sharply, wherein the stress–time curves exhibited sudden drops. As shown, the increase in axial stress consistently causes a subtle rise in both AE ringing counts and CE counts until both increase abruptly to reach to the peak at the same time and then drop together. Notably, the peak values of AE ringing counts show a little difference in the drying group. However, those under 8 MPa cell pressure are recorded as 25,805 and 14,710 in the pH = 2 group and 21,540 and 16,101 in the pH = 7 group at F-T cycles 0 and 40, respectively. Similarly, the peak values of CE counts at F-T cycles 0 and 40 are recorded as 857,419 and 392,537 in the pH = 2 group and 936,490 and 307,037 in the pH = 7 group, respectively. Similarly, the peak values of parameters decrease with the increase of F-T cycles for both pH groups subjected to the same D-W condition and cell pressure. As the F-T cycles increase from 0 to 40, the decrease of AE ringing counts in the pH = 2 and pH = 7 group is 11,095 and 5439, respectively. In essence, the order of reduction in the magnitude of AE ringing counts has been observed as: pH = 2 group > pH = 7 group > drying group. This could be attributed to the reason that the rock deterioration increased in the acidic environments due to the interaction of H+ and water. The former could accelerate the dissolution of calcite and feldspar from rock samples, given by the following chemical reactions:(6)$${\mathrm{CaCO}}_{3}{+2H}^{+}\to {\mathrm{Ca}}^{2+}{+\mathrm{CO}}_{2}{+H}_{2}O$$
(7)$${\mathrm{NaAlSi}}_{3}O_{8}{+4H}^{+}{+4H}_{2}O\to {\mathrm{Na}}^{+}{+\mathrm{Al}}^{3+}{+3H}_{4}{\mathrm{SiO}}_{4}$$
(8)$${\mathrm{KAlSi}}_{3}O_{8}{+4H}^{+}{+4H}_{2}O\to K^{+}{+\mathrm{Al}}^{3+}{+3H}_{4}{\mathrm{SiO}}_{4}$$

Not surprisingly, the peak values of both acoustic emission (AE) ringing counts and cumulative energy (CE) counts increase proportionately with the cell pressure. For instance, under the cell pressures of 4 MPa, 8 MPa, 16 MPa, and 32 MPa (at 40 F-T cycles), the observed AE values are 10,069, 14,710, 16,963, and 17,531, while CE values are 129,344, 392,662, 557,916, and 575,518, respectively. This could be attributed to the fact that the rock samples could store a higher amount of energy before failure at higher cell pressures indicated by the increased CE values (i.e., improved ductility) that is fully consistent with the increasing magnitude of energy released at failure.

### 3.4. The Characteristics of AE Positioning

Figure 9, Figure 10 and Figure 11 show AE positioning points and the corresponding rock failure diagrams under different environmental conditions and cell pressures simulated in this study. The AE positioning points could characterize cracks initiation and expansion due to F-T cycling (0 to 40 cycles) under different cell pressures (4 to 32 MPa). Before applying F-T cycling, there appears a few positioning points accompanied with some longitudinal cracks inside the rock samples. During the loading process, the microcracks slowly progress to develop shear expansion and slip, while the rock samples remain relatively intact, with a few fragments falling until failure in oblique shear. For up to 20 F-T cycles, the AE positioning points of the drying group are minimum and are more concentrated on the surface. The rock samples with powdery interior have the maximum AE positioning points and highest degree of fragmentation in the pH = 2 group. In the pH = 7 group, cracks continue to progress towards the ends from different directions to fully develop the failure. The number of AE positioning points and cracks increase consistently until 20 F-T cycles and there appears fragmentation at the tip of rock sample. Abundant secondary cracks deriving around the main crack gradually converge and eventually form macrocracks running through the rock surface. Due to incomplete initiation and expansion, there are both shear slip and extension along the tensile cracks. Thereon, the cracks become wider and deeper, while propagating from the surface to the interior, with the number of cracks increasing further at 40 F-T cycles. Both ends of a rock sample are pulverized in damage, and the number of strip and flake rock blocks on the surface increase substantially to induce tensile-oblique shear failure. In essence, at a given F-T cycle, the degree of internal deterioration is as follows (low to high): (drying group) > (pH = 7 group) > (pH = 2 group). 

As cell pressure increases under the same D-W and F-T conditions, the AE positioning points decrease and gradually concentrate near the surface of main fracture. Fracture diagram shows the decrease in the number of surface cracks. The distribution of AE positioning points is extensive and dispersed under 4 MPa cell pressure, while formation of small cracks and fragments lead to the fracture. Interestingly, under cell pressure of 32 MPa, there appear only macroscopic cracks, with AE positioning points concentrated around the point of failure.

**Figure 6 materials-16-04001-f006:**
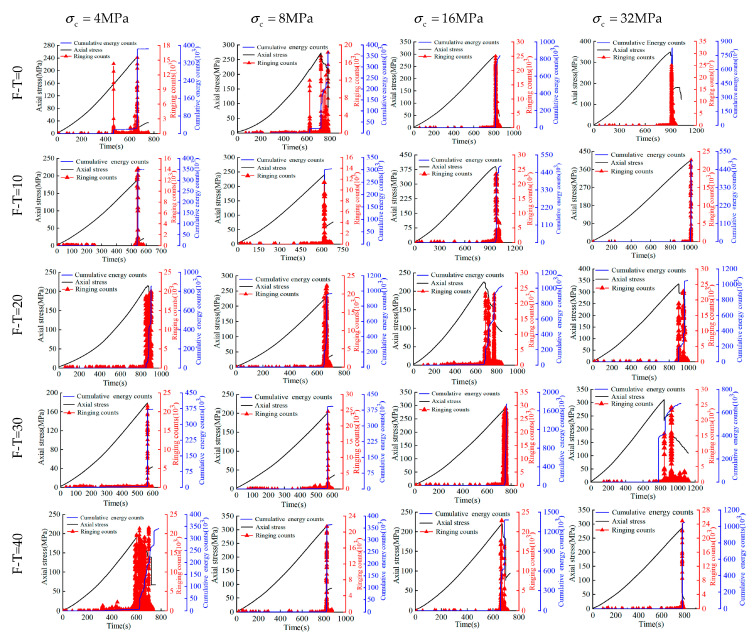
Triaxial compression stress–time–AE count curve of the drying group.

**Figure 7 materials-16-04001-f007:**
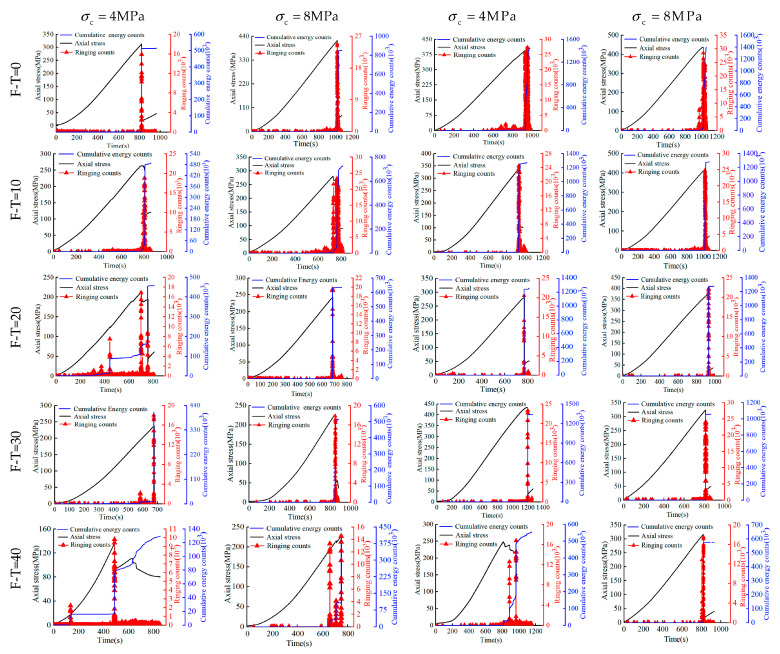
Triaxial compression stress–time–AE count curve of the pH = 2 group.

**Figure 8 materials-16-04001-f008:**
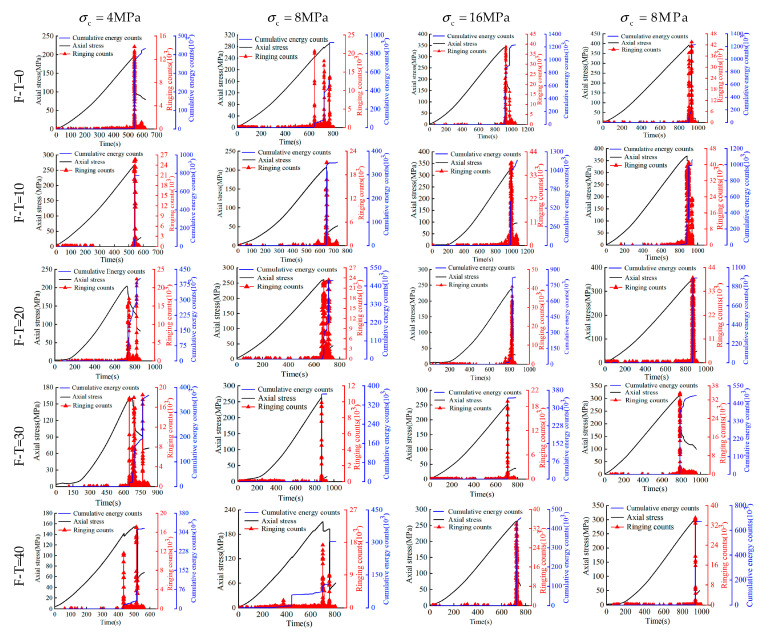
Triaxial compression stress–time–AE count curve of the pH = 7 group.

**Figure 9 materials-16-04001-f009:**
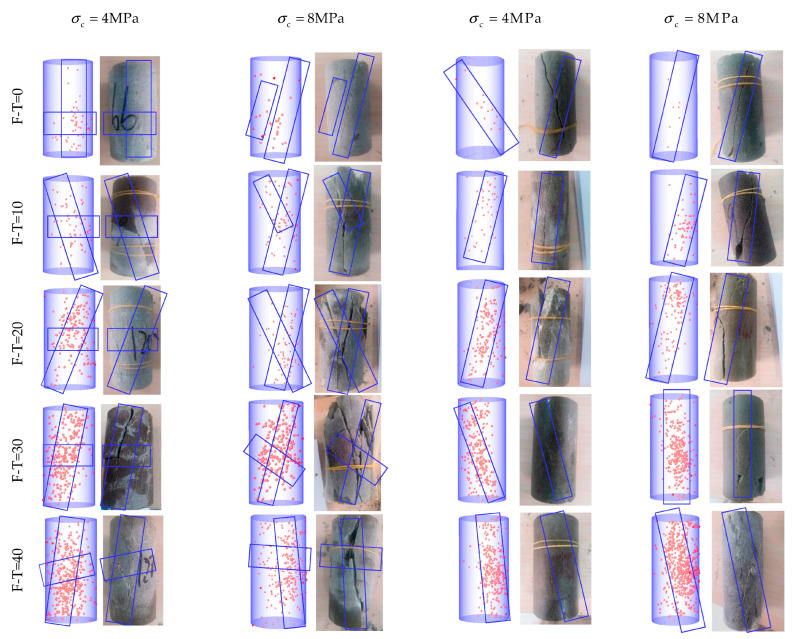
AE positioning points corresponding to the failure diagram of the drying group. (The red dots are AE positioning points and blue box are the crack propagation directions).

**Figure 10 materials-16-04001-f010:**
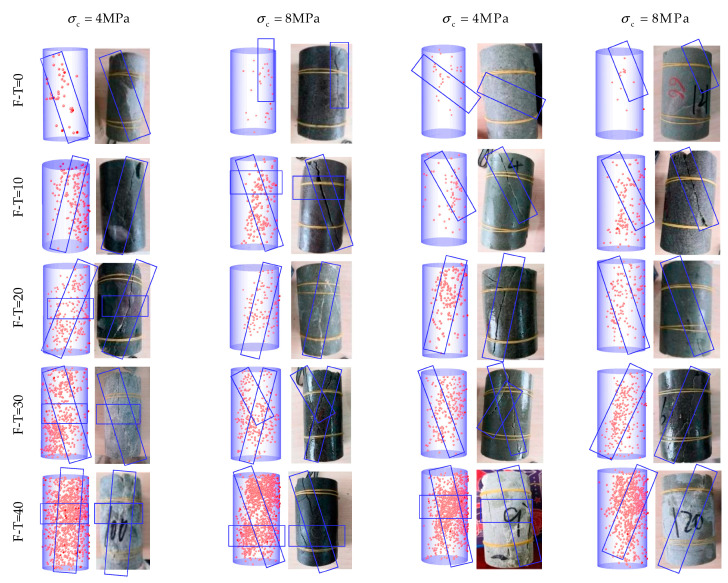
AE positioning points corresponding to the failure diagram of the pH = 2 group. (The red dots are AE positioning points and blue box are the crack propagation directions).

**Figure 11 materials-16-04001-f011:**
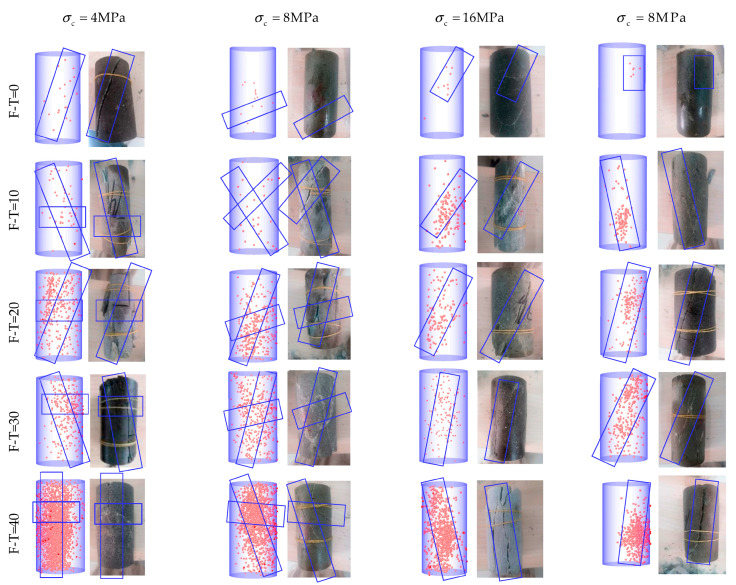
AE positioning points corresponding to the failure diagram of the pH = 7 group. (The red dots are AE positioning points and blue box are the crack propagation directions).

## 4. Damage Model

### 4.1. Cross-Scale Definition of Damage Variables

In this study, regardless of environmental and loading conditions, the cracks initiated and propagated until failure occurred that is termed as rock damage. According to the theory of damage mechanics, the damage variable D may be defined as:(9)$$D=1-\frac{E_{N_{t}}}{E_{0}}$$
where $E_{0}$ is the elastic modulus before F-T cycling and $E_{N_{t}}$ is the elastic modulus after $N_{t}$ freeze-thaw cycles.

The progressive failure behavior caused by the accumulation of microdefects within the rock under loading could be expressed by the damage variable defined by Kachanov [38]:(10)$$D_{n}=\frac{A_{d}}{A_{0}}$$
where $A_{d}$ is the cross-sectional area of the damaged part and $A_{0}$ is the cross-sectional area of the original rock sample.

The AE ringing counts for a unit area of the damaged rock sample would read [39]:(11)$$C_{w}=\frac{C_{0}}{A_{0}}$$
where $C_{0}$ is the AE ringing counts at the onset of complete damaged rock.

Similarly, for the rock sample damage area $A_{d}$, the AE ringing counts would read:(12)$$C_{d}=C_{w}A_{d}=C_{0}\frac{A_{d}}{A_{0}}$$

From Equations (10) and (12), the damage variable of rock may be given by:(13)$$D_{n}=\frac{C_{d}}{C_{0}}$$

During the test, the test machine often stopped working before the rock sample was completely damaged due to the performance of the test machine. Thus, the damage variable was less than 1. For reducing the error caused by this phenomenon, the correction factor $D_{U}$ is introduced [30]:(14)$$D_{U}=1-\frac{\sigma_{r}}{\sigma_{b}}$$
where $\sigma_{r}$ and $\sigma_{b}$ represent residual and peak strengths, respectively.

Subsequently, the damage variable $D_{n}$ takes the form:(15)$$D_{n}=(1-\frac{\sigma_{r}}{\sigma_{b}})\frac{C_{d}}{C_{0}}$$

According to the study of Zhang and Yang [29], the total damage variable $D_{m}$ may be expressed as:(16)$$D_{m}=D+D_{n}-DD_{n}$$

Now, using Equations (9) and (15) into Equation (16), the total damage variable may be written as:(17)$$D_{m}=1-\frac{E_{N_{t}}}{E_{0}}[1-\frac{C_{d}}{C_{0}}(1-\frac{\sigma_{r}}{\sigma_{b}})]$$

Previously, Cui et al. [36] established a damage variable based on porosity in order to examine the rock damage mechanism under both D-W and F-T conditions. They found that the change of damage variables in different solutions is significantly different, which is consistent with the results of this study. Similarly, Liu et al. [31] defined damage variables based on cumulative ringing counts and deduced the uniaxial strength of F-T rock. It was concluded that the cumulative ringing count could reflect the damage evolution process of F-T sandstone. Consequently, this current study defines the damage model based on both AE ringing counts and elastic modulus in tandem. The model can reflect the damage evolution and mechanism of Helankou rocks subjected to weathering and variable confining pressures. Notably, the effects of changing cell pressure have also been evaluated to replicate the site conditions, wherein the Helankou rocks are exposed to the natural environment. In this paper, the damage variable could be defined across scale by Equation (17) based on both the elastic modulus and AE ringing counts under the coupling of freeze-thaw-dry-wet cycling and variable loading conditions. In this way, it could define the rock damage caused by crack development and expansion due to both the increasing frost-heave forces and the loss of cementation in addition to the variable loading conditions. Consequently, the rock damage model under triaxial compression conforms to the generalized Hooke ‘s law and can be given by [38]:(18)$$\sigma_{1}=\frac{\lambda }{\lambda +G}\sigma_{c}+\frac{3\lambda G+{2G}^{2}}{\lambda +G}\varepsilon_{1}$$
where $\lambda =\frac{\mu E}{\left(1+\mu \right)\left(1-2\mu \right)}$, $G=\frac{E}{2(1+2\mu )}$, and μ is the Poisson ratio.

Substituting the Lame constant into Equation (18), the principal stress–strain constitutive relation based on rock damage may take the form:(19)$$\sigma_{1}=2\mu \sigma_{c}+E_{N_{t}}(1-D_{m})\varepsilon_{1}$$

Equation (19) satisfies the cross-scale definition of the damage model for Helankou rock materials.

### 4.2. Model Verification

To verify the proposed model, the experimental results of this study have been compared directly with the values obtained from the proposed model. The damage variables and the corresponding curves have been shown in Figure 12, Figure 13 and Figure 14 for the drying group, pH = 2 group, and pH = 7 group, respectively. As shown, under different F-T cycles after D-W cycling, theoretical results of stress–strain curves from the proposed model are fully consistent with the current experimental results. This clearly establishes that the stress–strain behavior of Helankou rocks can be captured through the proposed damage model based on both elastic modulus and AE ringing counts. Notably, the damage variables in all the figures remain low and constant initially, then rise to a maximum, and finally become stable. The peak value of the damage variable gradually increases with the increasing number of F-T cycles. The increasing range of damage variables increases, reaching a maximum at 20 F-T cycles, and then gradually decreases from 20 to 40 F-T cycles. For instance, under 8 MPa cell pressure for 0 to 20 F-T cycles, the damage variables of the drying group, pH = 2 group, and pH = 7 group have been noted as 0.226, 0.417, and 0.292, respectively. While damage variables for 20 to 40 F-T cycles computed for the drying group, pH = 2 group, and pH = 7 group are 0.005, 0.161, and 0.103, respectively. This discrepancy may be attributed to the fact that the initiation and propagation of cracks would be faster during 0 to 20 F-T cycles, and the corresponding magnitudes of damage variables would follow the descending order: pH = 2 group > pH = 7 group > drying group. Notably, the damage variables are smaller under higher cell pressures under same D-W and F-T conditions, thus indicating that the higher cell pressures may substantially retard the rate of deterioration in rocks. For example, under 8 MPa and 32 MPa cell pressures with 20 to 40 F-T cycles, the peak value of the damage variables is 0.462 and 0.418 for the drying group, 0.736 and 0.635 for the pH = 2 group, and 0.578 and 0.366 for the pH = 7 group, respectively. Nevertheless, the proposed model is a semi-empirical model, which has been tested against experimental data and found to be consistent. It could satisfactorily capture the experimental behavior of tested rock samples with higher accuracy and rigor. Thus, the proposed model may be adopted for preliminary assessments with enhanced confidence by the practitioners.

## 5. Scope and Limitations of This Study

This study tested a limited number of rock samples collected from the Helankou relics site in the Helan Mountains of Ningxia province in China. The current experiments have been carried out on fresh rock samples collected from the site with the specific focus on quantifying the impact of environmental factors on rock surface. As an obvious limitation associated to most experimental studies, the dimensions of current laboratory rock samples tested in this study may not be compared directly to those of most field problems. Additionally, the study did not run any in-depth investigations into geological and petrographic contents of rock mass tested, while the chemical analysis would also be warranted before directly implementing the outcomes of this study to field problems other than preliminary stages of planning and design. Nevertheless, while rigorous and intuitive approximation of the response for tested rock samples may be promptly obtained from the proposed damage model, the most accurate and reliable form of assessment would remain the specific field and laboratory testing to ascertain rock damage characteristics. In essence, the current study provides a rigorous framework for future research on rock damage modeling by coupling the impacts of environmental factors, aging due to physical phenomena, and chemical actions which have not been covered in this study. For this purpose, more specific data from both field and laboratory studies should be used in tandem to develop models with enhanced accuracy that would boost the confidence of practicing engineers.

## 6. Conclusions

This study experimentally examined the mechanical response of a select number of rock samples collected from the Helankou relics site in the Helan Mountains of Ningxia province in China. A series of freeze-thaw and wet-dry cycles have been implemented on tested samples to replicate weathering caused by extreme environmental conditions at the site. Subsequently, a series of triaxial compression and non-destructive tests at different cell pressures has been carried out to evaluate damage potential of rock samples. Accordingly, a semi-empirical damage model has been proposed that could successfully capture the response of currently tested rock samples. The specific findings from this study have been summarized as follows:➢The elastic modulus could then be correlated with a multivariate function, residual stress, and peak stress using a non-linear function decreasing with environmental deterioration (more F-T cycles) and increasing with the cell pressure. For instance, after 0 to 40 freeze-thaw cycles, the residual stress and peak stress in the pH = 2 group decreased by 146 MPa and 212 MPa, respectively. As the cell pressure increases from 4 to 32 MPa, the residual stress and peak stress in the drying group increased by 113 MPa and 169 MPa.➢The environmental effects on rock samples could be imitated in the laboratory, whereby freeze-thaw and dry-wet cycling could be done to simulate extreme weathers, while acidic, neutral, and basic environments could be modeled by varying the pH values of rock samples. For instance, cracking could be induced by F-T and D-W cycling, progressed, and propagated by the acidic environment, and controlled further through varying cell pressures.➢In this study, both acoustic energy and cumulative energy (CE) could capture the process of rock damage in the form of initiation and progression of cracks leading to the development of full failure. For instance, AE and CE ringing counts increased substantially at the onset of rock damage and dropped subsequently to indicate a marked strength reduction in rocks (i.e., development of rock failure).➢A new damage model has been proposed based on both elastic modulus and AE ringing counts that could capture the combined impacts of weathering and other mechanical agents damaging the rocks. The established model could successfully capture the damage evolution and mechanism of Helankou rocks under extreme climatic environmental conditions and high cell pressures. For a prompt and preliminary assessment, the rocks strength may be quantified through the proposed model that could provide basis for both risk assessment of rock paintings and technical support for preserving the rock relics in Helan Mountains region.

## Figures and Tables

**Figure 1 materials-16-04001-f001:**
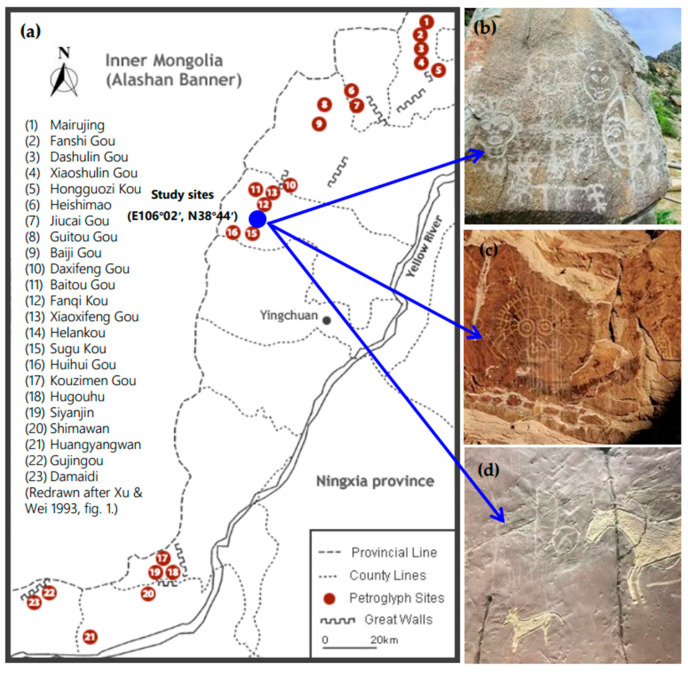
The position and rock damage of Helankou relics; (**a**) position [4], (**b**) powder peeling, (**c**) crack development, and (**d**) risk of collapse.

**Figure 2 materials-16-04001-f002:**
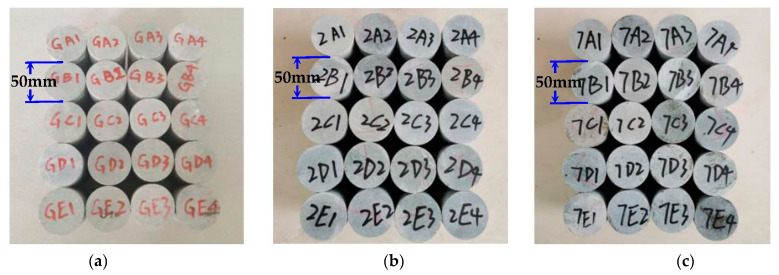
Specimens’ preparation: (**a**) drying group; (**b**) pH = 2 group; (**c**) pH = 7 group.

**Figure 3 materials-16-04001-f003:**
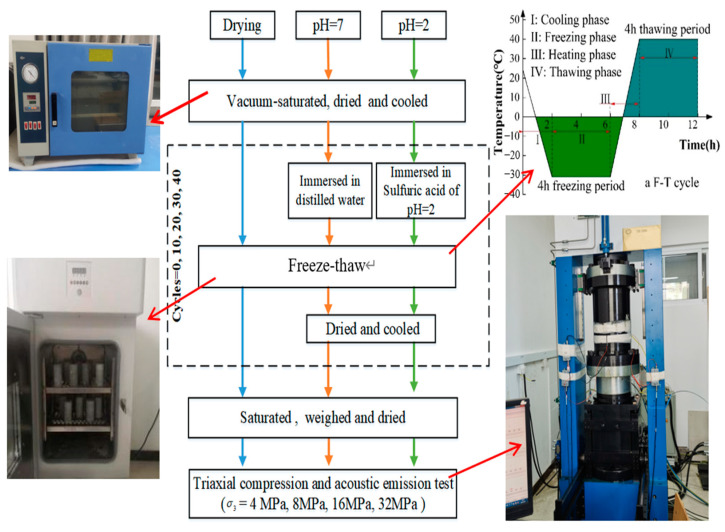
Flowchart of the experiments with the illustration of experimental setup.

**Figure 4 materials-16-04001-f004:**
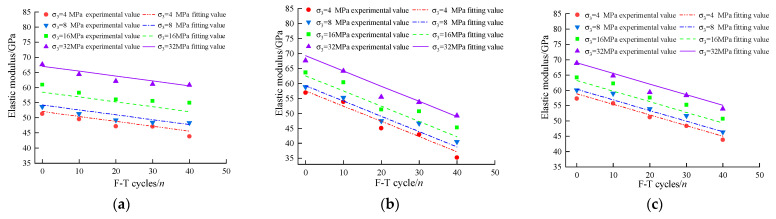
The fitting curves and experimental values of elastic modulus: (**a**) drying group, (**b**) pH = 2 group, (**c**) pH = 7 group.

**Figure 5 materials-16-04001-f005:**
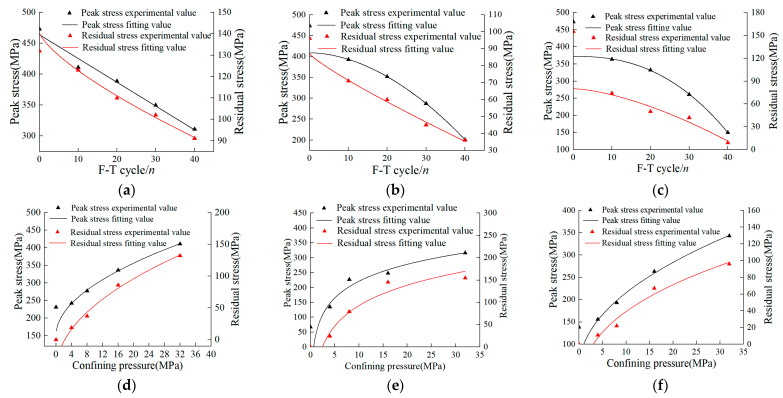
The fitting curves and experimental values under different F-T cycles and cell pressures: (**a**) drying group; (**b**) pH = 2 group; (**c**) pH = 7 group; (**d**) drying group; (**e**) pH = 2 group; (**f**) pH = 7 group.

**Figure 12 materials-16-04001-f012:**
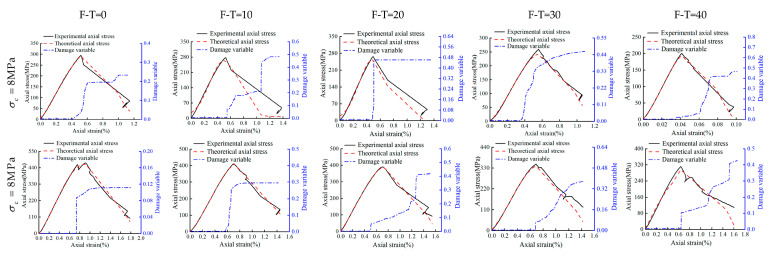
The damage variable, stress–strain experimental, and theoretical curve of the drying group.

**Figure 13 materials-16-04001-f013:**
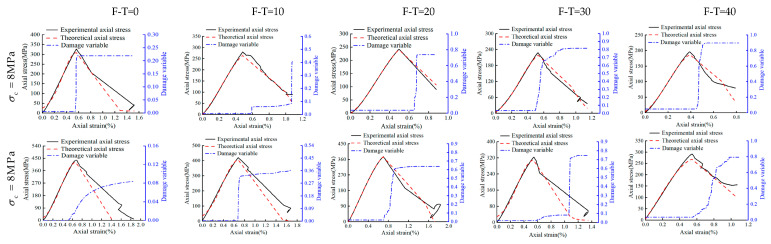
The damage variable, stress–strain experimental, and theoretical curve of the pH = 2 group.

**Figure 14 materials-16-04001-f014:**
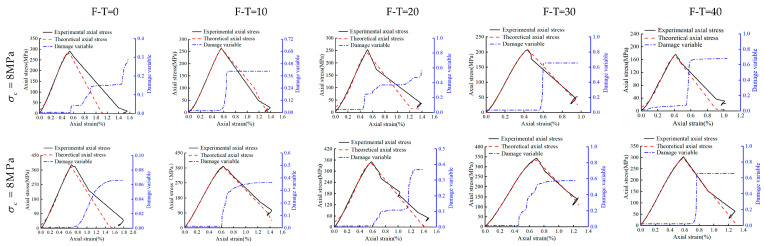
The damage variable, stress–strain experimental, and theoretical curve of the pH = 7 group.

**Table 1 materials-16-04001-t001:** Summary of model parameters adopted in this study.

Grouping	Drying Group	pH = 2 Group	pH = 7 Group
$a_{1}$/GPa	49.94	55.87	57.45
$b_{1}$	−0.16	−0.51	−0.35
$c_{1}$	0.53	0.42	0.36
R^2^	0.96	0.95	0.96
$a_{2}$/MPa	140.71	79.88	87.02
$b_{2}$	−3.36	−0.23	−2.15
$c_{2}$	0.73	1.55	0.86
R^2^	0.995	0.959	0.993
$a_{3}$/MPa	424.24	372.06	408.75
$b_{3}$	−0.31	−0.02	−0.21
$c_{3}$	1.60	2.44	1.87
R^2^	0.997	0.999	0.999
$a_{4}$/MPa	−51.88	−49,644.81	69.83
$b_{4}$	35.10	49,585.18	46.32
$c_{4}$	0.48	0.13 × 10^−2^	0.371
R^2^	0.990	0.936	0.971
$a_{5}$/MPa	149.07	−38,785.26	41.48
$b_{5}$	45.66	38,818.54	58.17
$c_{5}$	0.51	0.21 × 10^−2^	0.476
R^2^	0.999	0.949	0.998

## Data Availability

The data supporting the findings of this study are available from the corresponding author upon reasonable request.

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
