# Peer review of "A Semi-Empirical Damage Model of Helankou Rocks Based on Acoustic Emission"

_materials, 2023, doi:10.3390/ma16114001_

Round 1
Reviewer 1 Report
Dear authors. As you can see from my report, the paper is well written and interesting. An extensive update of the references is needed. 80% of the references are from China authors and this must be improved by more "international" bibliography. In the annex pdf document you can find some suggestions

Author Response
Point R1-1: An extensive update of the references is needed. 80% of the references are from China authors and this must be improved by more "international" bibliography.
Response R1-1: Thank you, we have now updated the revised manuscript with the following references:
7. Gratchev, I.; Pathiranagei, S.V.; Kim, D.H. Strength properties of fresh and weathered rocks subjected to wetting–drying cycles. Geomech. Geophys. Geo-energ. Geo-resour. 2019, 5, 211–221.
9. Mousavi, S.Z.S.; Tavakoli, H.; Moarefvand, P.; Rezaei, M. Assessing the effect of freezing-thawing cycles on the results of the triaxial compressive strength test for calc-schist rock. Int. J. Rock Mech. Min. 2019, 123, 104090.
12. Raynaud, S.; Fabre, D.; Mazerolle, F.; Geraud, Y.; Latière, H.J. Analysis of the Internal Structure of Rocks and Characterization of Mechanical Deformation by a Non-Destructive Method: X-ray Tomodensitometry. Tectonophysics 1989, 159, 149–159.
13.Hidajat, I.; Mohanty, K.K.; Flaum, M.; Hirasaki, G. Study of Vuggy Carbonates Using X-ray CT Scanner and NMR. In Proceedings of the SPE Annual Technical Conference and Exhibition, San Antonio, TX, USA, 29 September 2002.
15. Stanchits, S., Burghardt, J.; Surdi, A. Hydraulic Fracturing of Heterogeneous Rock Monitored by Acoustic Emission. Rock Mech Rock Eng. 2015, 48, 2513–2527.
18. Dornfeld, D.A.; Kannatey-Asibu, E. Acoustic emission during orthogonal metal cutting. J. Mech. Sci. 1980, 22(5), 285-296.
21. Hampton, J.; Gutierrez, M.; Matzar, L. Microcrack Damage Observations near Coalesced Fractures Using Acoustic Emission. Rock Mech Rock Eng. 2019, 52, 3597–3608.
23. Pirskawetz, S.M.; Schmidt, S. Detection of wire breaks in prestressed concrete bridges by Acoustic Emission analysis, Built Envir. 2023, 14, 100151.
Point R1-2: “which country?”
Response R1-2: “Ningxia, China” is added now (L17):
Point R1-3: “380C”
Response R1-3: “0C” has been changed to “°C”.
Point R1-4: about “acid rain and temperature changes”: I suggest here a work about the effects on rainwater at acid pH on limestone materials.
Sitzia, F. Lisci, C. Mirao, J. The interaction between rainwater and polished building stones for flooring and cladding - Implications in architecture. J. Build. Eng. 2022,52, 104495.
Response R1-4: Thank you for this suggestion, the following information has been added to the revised manuscript (L164-169):
“For brevity, drying groups were set up to study the effects of temperature changes (i.e., maximum temperature difference) on rocks, pH=7 groups were examined for quantifying water and temperature changes (e.g., rocks immersed in streams for long periods of time are affected by water and temperature changes), and pH=2 groups were considered for the effects of acid rain on rock materials [36].”
Point R1-5: why -31 to 40? are them temperature recorded in the ufficial climate data? To be complete, the papaer must indicate the climate condiction of the place like average min. average max. average annual temp and humidity and so on.
Response R1-5: In recent years, the average monthly maximum and minimum temperatures have been 29.9°C in July and -12.8°C in January respectively. In addition, according to the relevant records, the historical highest temperature reached 39 ℃ and lowest -31 ℃. The monthly average relative humidity is 40 % ~ 66 %. This information has been added to the revised paper (L53-L57).
Point R1-6: “these should be b5 and c5 or not?”
Response R1-6: Corrected (L241: Table 1).

Reviewer 2 Report
This study is very interesting and within the scope of the journal. However, the manuscript has a few important shortcomings that should be addressed in the revision.
1) In keywords: add “semi-empirical.”
2) In Figure 3: Kindly improve the image quality of left images and add the experimental set-up image instead of drawn image.
3) In Table 1: Heading should be above the Table.
4) In conclusion: Kindly add the experimental and analytical outcomes as bullet points. That will be more interesting for the reader.
5) Kindly add the future recommendations.
Author Response
Note: Line numbers of the revised manuscript where the corrections are made are indicated in bold letters and the corrections are also indicated with the relevant comment numbers on the left/ right margin of the marked copy. The comments are numbered for convenient perusal as R1-1 (Reviewer 1, point 1) etc.
Point R2-1: In keywords: add “semi-empirical.”
Response R2-1: Added to the revised manuscript (L38).
Point R2-2: In Figure 3: Kindly improve the image quality of left images and add the experimental set-up image instead of drawn image.
Response R2-2: Thank you, Figure 3 has been re-drawn with improved image quality, as suggested by this reviewer. Also, an image of the experimental setup has also been added.
Point R2-3: In Table 1: Heading should be above the Table.
Response R2-3: Corrected.
Point R2-4: In conclusion: Kindly add the experimental and analytical outcomes as bullet points. That will be more interesting for the reader.
Response R2-4: Done, we have now revised the conclusions in the light of this reviewer’s comments and updated the revised manuscript accordingly (L517-524):
“The elastic modulus could then be correlated with a multivariate function, residual stress and peak stress using a nonlinear function decreasing with environmental deterioration (more F-T cycles), and increasing with the cell pressure. For instance, after 0 to 40 freeze-thaw cycles, the residual stress and peak stress in pH = 2 group decreased by 146MPa and 212MPa, respectively. As the cell pressure increases from 4 to 32MPa, the residual stress and peak stress in drying group increased by 113MPa and 169MPa.”
Point R2-5: Kindly add the future recommendations.
Response R2-5: The future recommendations has been added in follows(L493-499):
“In essence, the current study provides a rigorous framework for future research on rock damage modelling by coupling the impacts of environmental factors, aging due to physical phenomena, and chemical actions that has not been covered in this study. For this purpose, more specific data from both field and laboratory studies should be used in tandem to develop models with enhanced accuracy that would boost the confidence of practicing engineers.”
Reviewer 3 Report
ARTICLE REVIEW
written by
Youzhen Yang, Qingqing Lin, Hailong Ma, Jahanzaib Israr, Wei Liu, Yishen Zhao, Wenguo Ma, Gang Zhang, and Hongbo Li
“A Semi-Empirical Damage Model of Helankou Rocks based
on Acoustic Emission”
submitted to “Materials”
This article is about building a semi-empirical Damage Model of Helankou Rocks based on Acoustic Emission.
The article is a completed study and is recommended for publication in the journal Materials. The results of using Acoustic Emission on unique objects represented by rock paintings are presented.
The following remarks are made:
1. It is necessary to indicate the lithological types of the investigated (destructed) rocks, their composition, structural and textural features.
2. There is no information on chemical weathering and the focus is on physical weathering. Specify the depth of impact of weathering on rocks.
3. It must be taken into account that at the time of drawing ancient drawings on the rocks, the rocks were already weathered.
4. It is desirable to remake the conclusion: make it more concise and contain concretely established (revealed) facts.
5. As an example of the study of various types of weathering, one can point to the article: Akulov, N.I.; Kashik, S.A.; Mazilov, V.N.; Fileva, T.S. Weathering crusts of the southern coast of Lake Baikal. Russian Geology and Geophysics. 1996, 37(10), 82-87.
Author Response
Note: Line numbers of the revised manuscript where the corrections are made are indicated in bold letters and the corrections are also indicated with the relevant comment numbers on the left/ right margin of the marked copy. The comments are numbered for convenient perusal as R1-1 (Reviewer 1, point 1) etc.
Point R3-1: It is necessary to indicate the lithological types of the investigated (destructed) rocks, their composition, structural and textural features.
Response R3-1: The following information has been added to the revised manuscript (L144-148):
“In essence, the mineral composition of the rock samples mainly consists of quartz (40 %), feldspar (22 %), calcite (18 %) and clay minerals (20 %). The angular-subangular quartz with a particle size of 0.15-0.9 mm, the subangular feldspar 0.18-0.65 mm and evenly distributed rock debris of 0.2-0.55 mm in the rock specimens.”
Point R3-2: There is no information on chemical weathering and the focus is on physical weathering. Specify the depth of impact of weathering on rocks.
Response R3-2: We apologize for the inconvenience caused by our way of writing in the original manuscript. However, it is pertinent to mention that this current study focuses on chemical induced weathering and its physical impact in the form of deteriorating Helankou relics that has been comprehensively described in section 3.3 of the original manuscript. For enhanced clarity, we have now updated the section heading in the revised manuscript reading “3.3 Chemical induced Variations of AE Parameters” (L289). It is noteworthy that the current experiments have been carried out on fresh rock samples collected from the site with the specific focus on quantifying the impact of environmental factors on rock surface. This information has now been added to the revised manuscript (L480-483).
Point R3-3: It must be taken into account that at the time of drawing ancient drawings on the rocks, the rocks were already weathered.
Response R3-3: We completely agree with this reviewer. Needless to mention that neither this current study assumes that the rocks were intact when these relics were drawn, nor does it aim at investigating the complex geological processes involved in rock formation. It is noteworthy that the current study focuses on quantifying the impact of the environmentally induced ongoing weathering processes and the subsequent damage of rock relics. This is now mentioned in the revised manuscript (L480-483).
Point R3-4: It is desirable to remake the conclusion: make it more concise and contain concretely established (revealed) facts.
Response R3-4: Thank you, we have now revised the conclusions in the light of this reviewer’s comments and updated the revised manuscript accordingly (L506-552).
Point R3-5: As an example of the study of various types of weathering, one can point to the article: Akulov, N.I.; Kashik, S.A.; Mazilov, V.N.; Fileva, T.S. Weathering crusts of the southern coast of Lake Baikal. Russian Geology and Geophysics. 1996, 37(10), 82-87.
Response R3-5: Thank you for suggesting this reference, which is now cited in the revised paper (L66-67).